# Association of Organizational Behavior with Work Engagement and Work-Home Conflicts of Physician in China

**DOI:** 10.3390/ijerph18105405

**Published:** 2021-05-19

**Authors:** Nannan Liu, Yimei Zhu, Xiaoyu Wang, Hongwei Jiang, Yuan Liang

**Affiliations:** 1Department of Social Medicine and Health Management, School of Public Health, Tongji Medical College, Huazhong University of Science and Technology, 13 Hangkong Road, Wuhan 430030, China; liunannan301@163.com (N.L.); wangxiaoyu361@163.com (X.W.); 2School of Media, Communication and Sociology, University of Leicester, Leicester LE1 7RH, UK; yz411@leicester.ac.uk; 3Department of Epidemiology and Health Statistics, Tongji Medical College, Huazhong University of Science and Technology, 13 Hangkong Road, Wuhan 430030, China; jhwccc@sina.com

**Keywords:** organizational behavior, work engagement, work-home conflicts, physician

## Abstract

This study aimed to examine how organizational behavior is associated with work engagement (WE) and work-home conflicts (WHCs) of physicians. The data were from a national cross-sectional survey of 3255 Chinese physicians. We examined organizational fairness, leadership attention, and team interaction for organizational behavior. The results indicate that greater organizational fairness is associated with higher WE and lower WHCs. High task fairness was associated with greater pride, and more enjoyment in work, lower sense of guilt towards their family, and less complaints from family members. Physicians reporting higher levels of leaders’ attention to their opinions reported experiencing more enjoyment of their work, and less effects on their care for family. A greater number of dinners with colleagues per month was associated with higher WE and lower WHCs, whilst a greater number of clinical case meetings per month was associated with higher WE and higher WHCs. The results suggest that the behavior of organizations could be an important intervention to improve the wellbeing of physicians.

## 1. Introduction

Satisfactory working and living conditions for physicians are not only important for physicians’ own health and professional development, but also important for the quality of medical services and patients’ safety [1,2,3,4,5]. Over the past several decades, the social and working environments involved in medical practice have undergone significant changes, and physicians are facing increasing pressure from both work and everyday life [6,7]. Physicians appear to be getting busier, more tired, and more likely to neglect their families, while patients appear to be increasingly dissatisfied. As a result, the doctor–patient relationship seems to be increasingly unfriendly [8,9,10].

Faced with the prevalence of high work and life pressures, the focus of recent studies has shifted from individual factors to organizational factors, including working conditions/environments, work process improvement, occupational safety climate, workplace norms, leadership, and organizational support [11,12,13]. Although these organizational factors are related to physicians work and life pressures to a certain extent, including increasing work engagement (WE) and reducing work-home conflicts (WHCs), those findings offer limited guidance to hospital administrators and policy makers. This is due to the fact that work environment-related factors involve multiple subjects, such as hospitals, departments, clinicians, and patients, they are broad and scattered, and can be difficult to understand for hospital administrators and policy makers [12,13,14,15]. Furthermore, some factors, such as duty roster changes and working overtime, may be a norm for physicians since they need to prioritize care for patients, and are often subject to little modification. Therefore, the focus of predictors or potential possible targets not only needs to be easily modified, but requires a clear carrier and specific content [12,14,15,16,17]. One example is organizational pay equity, with hospitals or department teams (not clinicians or patients) as the carrier, and salary policy as the specific content. Accordingly, the research focus may need to shift from organizational factors to organizational behavior, such as organizational fairness, leadership behavior, and team interaction [14,15,16,17]. In this scenario, the carrier is organizations (hospitals or department teams or organizational legal team/leadership), and the content is behavior (such as organizational fairness, leadership support or attention, and team interactions). However, this idea has received little study to date. The effects of organizational behavior in healthcare are not only large-scale, but also multi-dimensional. Organizational behavior has been largely overlooked for a wide range of types of distress among physicians. Although some previous research has focused on leadership behavior, these studies have not extensively adjusted for organizational structural factors (such as hospital nature, level or teaching status) and patient factors, many of which confound work and life pressures for clinicians.

Many studies have recognized that it is important to have employees who are engaged in their work. Thus, work engagement, defined as a positive, fulfilling, work-related state of mind that is characterized by vigor, dedication, and absorption, has become an important issue in the past few years [18,19,20]. Studies on the influencing factors of work engagement, in addition to personal factors, the most discussed included job resources, perceived supports, learning organizations, and transformational leadership. Despite this, little research has been found that explored the potential correlations between multiple organizational behaviors and work engagement among the physician [21,22]. On the other hand, with the increase of double-income families, work–family conflict, defined as a form of inter-role conflict in which the role pressures from the work and family domains are mutually incompatible in some respect, has become a prominent societal concern, attracting growing attention from researchers and policy makers [23]. Although some studies have focused on the impact of organizational behavior on employees’ work-family conflict, their participants are mainly enterprise employees, engineering and technical personnel and teachers [24], less with physicians.

The aim of this current study is to examine how organizational behaviors are associated with WE and WHCs among physicians, adjusting for demographic characteristics, hospitals and department characteristics, family support and patient behavior characteristics using data from a national physician survey. We hypothesized that physicians who reported positive organizational behavior are more likely to show better work engagement and less work-home conflicts than those who reported negative organizational behavior.

## 2. Methods

### 2.1. Study Design and Participants

This study is based on a stratified cluster sampling survey conducted across the whole of mainland China. The details of this survey have been described in a previous report [10]. Briefly, we selected six provinces (Gansu, Yunnan, Jiangsu, Shandong, Hubei, and Guangdong) and metropolitan Beijing, China’s capital, and there was a total of 85 eligible hospitals, of which 8 refused to participate, leaving a total of 77 participating hospitals (90.59%) [25]. A total of 528 departments were involved and all full-time physicians in the 528 departments were eligible to complete the survey. Participants provided oral informed consent for interviews. There were 5754 eligible respondents, of whom 1473 did not complete the survey (25.60%). We excluded 634 (11.02%) invalid questionnaires that contained errors or erratic responses after three trained research assistants conducted a manual check for handwriting and a computer-assisted quality assurance check during data entry. We also excluded 392 (6.81%) responses with missing key variables, such as dependent variables sex and age. The final analysis used data from 3255 (56.57%) remaining responses (see Appendix A).

We obtained ethics approval from the review board of the authors institute (no. IORG0003571).

### 2.2. Measures

#### 2.2.1. Exposure Factor

Exposure factors included three aspects of organizational behavior: Organizational fairness, leadership attention, and team interaction. The first two of these variables represent top-down relationships in organizational behavior, while the latter represents parallel relationships. Although there are several commonly used questionnaires for organizational fairness, leadership attention, and team interaction, such as Colquitt’s Organizational Justice Scale, the length of these questionnaires limits their feasibility for nationwide studies [5,6,26]. In the current study, organizational justice was assessed using two single-item measures adapted from the full Colquitt’s Organizational Justice Scale [27]: Pay equity (reflecting distributional fairness), and task fairness (reflecting procedural fairness). Leadership was assessed using two single-item measures: Interests attention (reflecting leadership’s attention for physicians’ material needs) and opinions attention (reflecting leadership’s attention for physicians’ spiritual needs). Team interaction was assessed using two single-item measures: Number of dinners with colleagues per month (reflecting social interactions) and number of clinical case meetings per month (reflecting work interaction) (for specific questions, see Appendix A).

#### 2.2.2. Outcome Variables

Outcomes included WE (reflecting the impact on work) and WHCs (reflecting the impact on family life). Although the Utrecht Work Engagement Scale (UWES) is considered the reference standard for the assessment of engagement, including vigor, dedication, and absorption, its length (22 items) limits its feasibility in a nationwide study. In prior studies, evaluation of work engagement has focused on the presence of vigor and dedication as the cornerstones of work engagement [18,28]. Therefore, in the current study, WE was assessed using two single-item measures adapted from the UWES. Pride in work was assessed using the question, “In the past year, to what extent have you felt proud of your work?” and enjoyment of work was assessed using the question “In the past year, to what extent have you enjoyed your work?”. This instrument measured overall WE on a scale of 1 to 5, and high WE was defined as responses in the highest two categories of this item. The outcome variables were recoded into binary variables—very low, somewhat low, and neutral were recoded to zero; somewhat high and very high were recoded to 1.

WHCs were assessed using three items: The effect on physicians’ family life was assessed using the question “In the past year, to what extent has your busy work schedule affected your ability to care for your family?”; physicians’ sense of guilt regarding their family was assessed using the question “In the past year, to what extent have you felt guilty towards your family due to the busy work schedule?”; and complaints from physicians’ families were assessed using the question “In the past year, to what extent have your family members complained due to the busy work schedule?” (response options: Very low, low, neutral, high, very high). Individuals who indicated very high or high were considered to exhibit WHCs, whereas those who indicated very low or low or neutral were considered to be satisfied with their work-home balance. In this study, the Cronbach’s α coefficients for WE and WHCs were 0.813 and 0.632, respectively (Appendix A).

#### 2.2.3. Control Variables

We also measured a number of factors previously shown to be associated with WE and WHCs among physicians, including sociodemographic factors (such as sex, age, marital status, education level), hospital and departmental characteristics (such as hospital level, hospital type, academic status), and family and patient behavior characteristics (such as family support, patient trust). Notably, in China, doctoral education in the medical profession only started in the 1980s. Hence, very few senior physicians have doctorates. Medical education in China is offered at the undergraduate level and below, such as junior college level and technical secondary school level (the latter two mainly serve rural and poor areas).

#### 2.2.4. Statistical Analysis

Data were weighted to adjust for non-responses so that participants responding to the initial questions matched the demographic characteristics of the total hospital staff population issued by the National General Hospital in 2015. We performed binary logistic regressions to evaluate associations between organizational behaviors, WE and WHC. All of the models were adjusted for physicians’ sociodemographic characteristics, hospital and department characteristics, and physicians’ family and patient behavior characteristics, which have previously been associated with WE and WHCs. All of the tests were two-sided, and p-values of 0.05 or less were considered statistically significant. All analyses were performed using SPSS, version 22.0 (SPSS Inc., Chicago, IL, USA).

#### 2.2.5. Sensitivity Analysis

We also conducted a sensitivity analysis of all models to examine the robustness of our findings. We re-specified our model and additionally adjusted for three WHC factors to WE. We also correspondingly adjusted for two WE factors to WHCs, in addition to all of the sociodemographic characteristics, hospital and department characteristics, and physicians’ family and patient behavior characteristics listed above.

## 3. Results

The characteristics of physicians are summarized in Table 1. The results revealed that 36.35% of study participants were aged 45 or older and 56.58% were male. Participants with PhD qualifications accounted for 11.47% of the sample, participants affiliated with West Hospital accounted for 72.22% of the sample, participants from tertiary hospitals accounted for 84.94% of the sample, and participants from teaching hospitals accounted for 19.68% of the sample. Similar proportions of participants were involved in internal medicine and surgery (52.94% vs. 47.06%, respectively).

Table 2 shows the distribution of WE and WHCs among physicians. Overall, physicians’ WE was reported to be dissatisfactory, only 17.57% of participants reported being proud of their work, and only 15.54% of participants reported enjoying their work. Substantial differences in WE were observed with differences in organizational behavior (Figure 1). In general, with increased positive organizational behavior, WE (both pride in work and pleasure in work) showed an increasing trend, and the difference was significant (*p*-value < 0.001). Differences in WHCs were also observed with changes in organizational behavior (Figure 2). In general, except for case discussions, with increased positive organizational behavior, WHCs, including the effect of work commitment on physicians’ care for families, the sense of guilt towards their families, and complaints from their families, showed a downward trend.

Table 3 summarizes the adjusted results of organizational behavior with WE and WHCs. We performed multivariable analysis to identify factors independently associated with WE or WHC. In general, greater organizational fairness was associated with higher WE and lower WHCs, and the significant effects of task fairness were clearer. Compared with those who rated “very bad” for task fairness, respondents who rated “somewhat good or very good” were more likely to report greater pride in work (OR = 2.37, 95% CI: 1.35–4.18), and more pleasure in work (OR = 2.64, 95% CI: 1.39–5.01), lower sense of guilt towards family (OR = 0.45, 95% CI: 0.26–0.77), and less complaints from family members (OR = 0.52, 95% CI: 0.35–0.77). In general, greater leaders’ attention to opinions was associated with higher WE, and lower WHCs. Compared with those who rated “very bad” for leaders’ attention to opinions, those who rated “somewhat good or very good” reported more pleasure in work (OR = 1.86, 95% CI: 1.07–3.26), and less effects on their care for family due to work commitment (OR = 0.64, 95% CI: 0.42–0.96). A greater number of dinners with colleagues in the past month was associated with higher WE and lower WHCs. However, a greater number of clinical case meetings in the past month was associated with higher WE and higher WHCs. Compared with those who reported 0–1 clinical case meetings in the past month, those who reported 4 or more clinical case meetings in the past month were 1.74-times more likely to affect their care for family (95% CI: 1.40–2.16), 1.98-times more likely to feel guilty towards their family (95% CI: 1.51–2.59), and 1.27-times more likely to receive complaints from family members (95% CI: 1.02–1.58). The full results of the models are available in Appendix A.

In our sensitivity analysis of WE (additionally adjusted for WHCs) and WHCs (additionally adjusted for WE), the findings remained largely the same (see Appendix A).

## 4. Discussion

This is the first study that provides national empirical evidence with regards to the association between organizational behaviors and physicians’ WE and WHCs in China. Overall, low levels of WE and high levels of WHCs were found to be common among physicians in China, with less than 20% of physicians reporting WE and the vast majority felt guilty towards their families due to their busy work schedule. Furthermore, the current results revealed that positive organizational behavior factors, including organizational fairness, leadership attention, and team interaction, were generally associated with increased WE and reduced WHCs. These findings indicate that organizations may improve the mental state of medical staff, by improving relevant organizational behaviors.

The findings of this study make empirical contributions to the current knowledge in medical practitioners’ mental wellbeing research field, which has received international attention in recent years. The increasing level of stress in relation to work and life balance among physicians indicates that research attention needs to move from the individual susceptibility (individual perspective), to the environment (group perspective) in order to consider for interventions from the organization’s perspective. Environmental factors affecting the mental state of physicians involve many aspects, including behaviors of physicians, their patients, their families (representing the social environment outside the hospital), and healthcare provider organizations, as well as low-variability characteristics inherent in medical work. Accordingly, intervention targets related to individual physicians’ or patients’ behaviors can be difficult to identify and too complicated to measure [3,29,30,31]. Hence, the intervention object investigated for this study is transformed from the individual to the group (organization).

In the context of environmental changes, both in social (e.g., market-based competition and improvement in patient requirements) and work contexts (e.g., new technology-based learning, research requirements, medical-related paperwork), individual doctors are often powerless, which makes the role of the hospital organization particularly important [32,33]. In the face of so many changes in the external environment, hospital organizations are required to make corresponding changes. Furthermore, given the high levels of work and life distress among physicians, effective intervention objects may be less focused on behaviors of individual physicians (e.g., methods for handling stress, enhancing professional ethics, self-improvement, and the spirit of service), and more focused on behaviors of the healthcare provider organization (e.g., improving organizational fairness, leadership behavior, and creating conditions and opportunities to promote team interaction) [32,34,35]. This transformation in organization behaviors can be an effective approach for enhancing work enjoyment and improving the work and life balance of medical practitioners.

The causal mechanisms underlying the current findings are uncertain, and our study was not designed to address this issue. However, it may be unsurprising that a high level of organizational fairness was associated with high WE, given previous research suggesting that low organizational fairness is associated with organizational rejection and a reduced sense of belonging in the organization, in addition to reduced dignity among staff [11,36]. The same mechanism may largely explain the association we observed between high levels of leaders’ attention to physicians’ material and spiritual needs and their WE and WHCs.

In addition, two important points should be noted. First, prior research indicates that, compared with distribution fairness, procedural fairness has a clearer impact on employees’ organizational citizenship behavior and psychological security [37,38]. The current results indicate that increased pay equity (reflecting fairness of distribution) and task fairness (reflecting procedural fairness) were associated with a trend for increased WE. However, in terms of statistical significance, compared with pay fairness, the impact of task fairness on WE was clearer, which to some extent confirms and extends the impact of distributional fairness and procedural fairness on employee work and mental health.

Second, compared with organizational fairness, leaders’ attention had a weaker impact on employee WE and WHCs, which could be related to two possible explanations. The first potential explanation involves the nature of these two types of organizational behavior. While organizational fairness and leadership attention are top-down relationships, organizational fairness focuses on institutionalism, whereas leadership attention is based on subjectivity (individual leaders) [12,15,17,39,40]. We argue that this relationship has similarities to the nature of football matches, with organizational fairness being similar to competition rules, and leadership attention being similar to the behavior of the referee. For players, the importance of fair game rules may be greater than the referee’s personal behavior. The other factor is related to the nature of physicians’ work, and particularly the autonomy associated with it. Physicians are able to start their own practice, and can practice at multiple hospitals and clinics. Therefore, the influence of leader attention may be reduced.

This study has a number of limitations. First, the response rate of the sample was relatively low. Due to financial constraints, we did not use material compensation to improve response rates (although, it should be noted that material compensation can potentially increase selection bias and measurement bias due to temptation). Low response rates are common in clinician surveys worldwide [41], particularly in China. To address this limitation, we used a sample of doctors that corresponded to the national demographic statistics in the same period as the survey to weight the sample and improve its representativeness. Second, the cross-sectional design made it impossible to infer a causal link between the study variables, and subsequent intervention studies may be valuable. Third, due to cultural differences between countries, understandings of WE and WHCs may differ between populations, and the measurement tools used in different studies have not been consistent [5,27], potentially limiting the generalizability of the current results to other countries.

## 5. Conclusions

In a nationally representative sample, low levels of WE and high levels of WHCs among physicians were found to be common in China. The results revealed that positive organizational behavior, particularly task fairness (reflecting procedural fairness) and the number of meals with colleagues (reflecting team social interactions) were associated with significantly increased WE and reduced WHCs, and our hypothesis has been confirmed. Compared with environmental and organizational factors, organizational behavior is not only more modifiable, but also more specific and actionable. If a causal relationship between organizational behavior and physicians’ WE and WHCs exists, given the increasing prevalence of physicians’ distress, the appropriate target of interventions would be no longer the behavior of physicians, but the behavior of organizations. Focusing on organizational behavior changes may make it possible to increase WE and reduce WHCs on a large scale, potentially providing an effective approach for mass prevention and treatment for improving physicians’ well-being.

## Figures and Tables

**Figure 1 ijerph-18-05405-f001:**
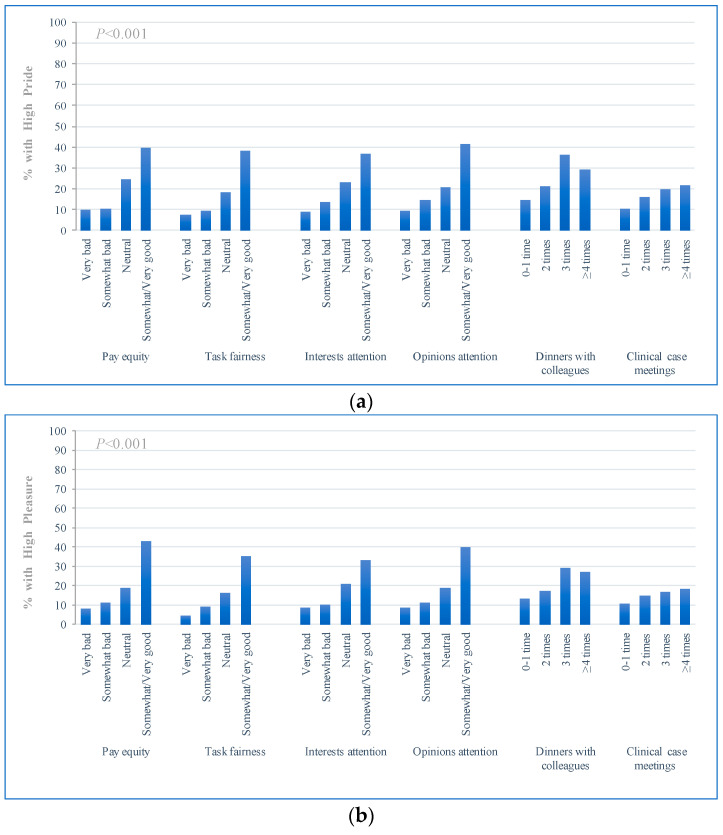
Organizational behavior with work engagement of physician. (**a**) High pride; (**b**) high pleasure.

**Figure 2 ijerph-18-05405-f002:**
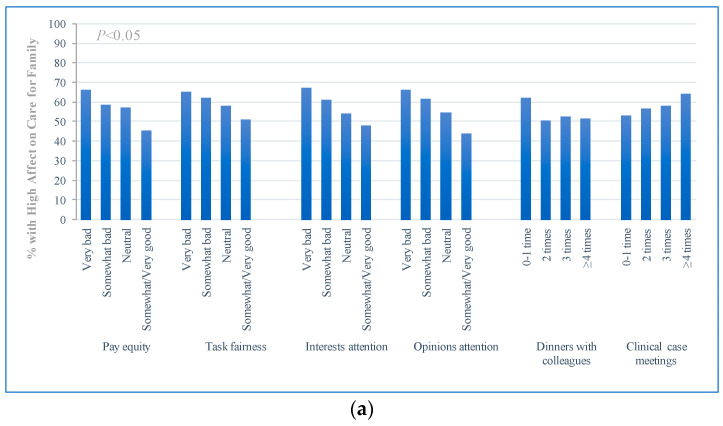
Organizational behavior with work-home conflicts of physician. (**a**) High effect on care for families; (**b**) high guilt towards families; (**c**) high complaints from families.

**Table 1 ijerph-18-05405-t001:** Sociodemographic characteristics of physician.

Characteristic	*N*(%; 95% CI)
**Socio–Demographic Characteristics**	
Sex	
Men	1842(56.58; 54.88–58.28)
Women	1413(43.42; 41.72–45.12)
Age, y	
≤34	945(29.02; 27.46–30.58)
35–44	1127(34.63; 33.00–36.26)
≥45	1183(36.35; 34.70–38.00)
Marital status	
Single/other	515(16.48; 15.21–17.75)
Married	2609(83.52; 82.25–84.79)
Education level	
Undergraduate and below	1679(52.93; 51.22–54.62)
Masters	1130(35.61; 33.96–37.26)
PhD	364(11.47; 10.38–12.56)
Economic status	
Very bad	421(12.96; 11.81–14.11)
Somewhat bad	653(20.12; 18.74–21.50)
Neutral	1903(58.63; 56.94–60.32)
Good	269(8.28; 7.33–9.23)
Title	
Primary/other	695(24.22; 22.75–25.69)
Middle	907(31.63; 30.03–33.23)
High	1267(44.15; 42.44–45.86)
**Hospital and Departmental characteristics**	
Hospital level	
Secondary	490(15.06; 13.83–16.29)
Tertiary	2765(84.94; 83.71–86.17)
Hospital type	
Traditional Chinese medicine	904(27.78; 26.24–29.32)
Western medicine	2351(72.22; 70.68–73.76)
Academic status	
Nonteaching	2615(80.32; 78.95–81.69)
Teaching	641(19.68; 18.31–21.05)
Physician specialty	
Internal medicine	1723(52.94; 51.23–54.65)
Surgery	1532(47.06; 45.35–48.77)
The ratio of physicians to beds	
<0.20	932(28.62; 27.07–30.17)
0.20–0.30	1304(40.05; 38.37–41.73)
≥0.30	1020(31.34; 29.75–32.93)
**Family support**	
Very low/Somewhat low	115(3.52; 2.89–4.15)
Neutral	574(17.66; 16.35–18.97)
Somewhat high/Very high	2564(78.82; 77.42–80.22)
**Patient behavior**	
Patient trust	
Very low/Somewhat low	1375(42.51; 40.81–44.21)
Neutral	1496(46.24; 44.53–47.95)
Somewhat high/Very high	364(11.26; 10.17–12.35)
Unreasonable request from the patient	
Very low/Somewhat low	1047(32.22; 30.61–33.83)
Neutral	1102(33.89; 32.26–35.52)
Somewhat high/Very high	1102(33.89; 32.26–35.52)

**Table 2 ijerph-18-05405-t002:** Work engagement and work-home conflicts of physician.

Variable Description	*N*(%; 95% CI)	Recategorization
**Work engagement**		
Pride		
Very low	594(18.23; 16.90–19.56)	No
Somewhat low	751(23.08; 21.63–24.53)	No
Neutral	1339(41.12; 39.43–42.81)	No
Somewhat high	478(14.69; 13.47–15.91)	Yes
Very high	94(2.88; 2.31–3.45)	Yes
Pleasure		
Very low	712(21.87; 20.45–23.29)	No
Somewhat low	922(28.34; 26.79–29.89)	No
Neutral	1115(34.25; 32.62–35.88)	No
Somewhat high	432(13.27; 12.10–14.44)	Yes
Very high	74(2.27; 1.76–2.78)	Yes
**Work–home Conflicts**		
Affecting care for family		
Very low	408(12.53; 11.39–13.67)	No
Somewhat low	422(12.96; 11.81–14.11)	No
Neutral	500(15.37; 14.13–16.61)	No
Somewhat high	869(26.69; 25.17–28.21)	Yes
Very high	1056(32.44; 30.83–34.05)	Yes
Guilty towards family		
Very low	49(1.51; 1.09–1.93)	No
Somewhat low	149(4.57; 3.85–5.29)	No
Neutral	532(16.33; 15.06–17.60)	No
Somewhat high	1184(36.37; 34.72–38.02)	Yes
Very high	1342(41.23; 39.54–42.92)	Yes
Complaint from family		
Very low	141(4.32; 3.62–5.02)	No
Somewhat low	440(13.52; 12.35–14.69)	No
Neutral	951(29.21; 27.65–30.77)	No
Somewhat high	976(29.99; 28.42–31.56)	Yes
Very high	747(22.96; 21.52–24.40)	Yes

**Table 3 ijerph-18-05405-t003:** Multivariable logistic regression results for correlates of organizational behavior-related effects.

Organizational Behavior	%(95% CI)	Work Engagement		Work–Home Conflicts		
		High Pride (OR, 95% CI)	High Pleasure (OR, 95% CI)	High Effect on Care for Family (OR, 95% CI)	High Guilt towards Family (OR, 95% CI)	High Complaints from Family (OR, 95% CI)
**Organizational fairness**						
Pay equity						
Very bad	27.88(26.34–39.42)	1[reference]	1[reference]	1[reference]	1[reference]	1[reference]
Somewhat bad	27.74(26.20–29.28)	0.55(0.36–0.83)	0.84(0.56–1.28)	0.84(0.65–1.08)	0.66(0.47–0.94)	0.72(0.56–0.92)
Neutral	36.50(34.85–38.15)	1.05(0.70–1.57)	0.93(0.61–1.42)	0.96(0.73–1.27)	0.66(0.45–0.95)	0.68(0.52–0.89)
Somewhat good/Very good	7.89 (6.96–8.82)	0.75(0.43–1.33)	1.32(0.75–2.32)	0.65(0.43–1.00)	0.38(0.23–0.63)	0.38(0.24–0.58)
Task fairness						
Very bad	15.11(13.88–16.34)					
Somewhat bad	21.51(20.10–22.92)	1.01(0.60–1.69)	1.68(0.93–3.04)	1.01(0.75–1.37)	0.48(0.30–0.77)	0.63(0.46–0.87)
Neutral	49.93(48.21–51.65)	1.27(0.78–2.07)	2.12(1.20–3.74)	1.03(0.77–1.39)	0.48(0.30–0.76)	0.48(0.35–0.66)
Somewhat good/Very good	13.46(12.29–14.63)	2.37(1.35–4.18)	2.64(1.39–5.01)	0.98(0.67–1.44)	0.45(0.26–0.77)	0.52(0.35–0.77)
**Leadership attention**						
Interests attention						
Very bad	31.35(29.76–32.94)					
Somewhat bad	23.76(22.30–25.22)	1.01(0.64–1.59)	0.70(0.43–1.14)	0.76(0.56–1.01)	1.03(0.70–1.51)	0.97(0.72–1.30)
Neutral	35.82(34.17–37.47)	1.22(0.77–1.92)	1.09(0.68–1.74)	0.77(0.57–1.05)	0.86(0.58–1.27)	0.87(0.64–1.18)
Somewhat good/Very good	9.07 (8.08–10.06)	1.96(1.12–3.45)	1.23(0.69–2.21)	0.68(0.45–1.03)	1.18(0.70–1.99)	0.81(0.53–1.24)
Opinions attention						
Very bad	33.30(31.68–34.92)					
Somewhat bad	24.18(22.71–25.65)	1.07(0.70–1.63)	0.97(0.62–1.52)	1.18(0.89–1.56)	1.17(0.81–1.69)	0.97(0.73–1.28)
Neutral	32.88(31.27–34.49)	1.11(0.71–1.74)	1.27(0.80–2.02)	0.87(0.65–1.18)	1.03(0.70–1.53)	1.05(0.78–1.43)
Somewhat good/Very good	9.63(8.62–10.64)	1.40(0.81–2.43)	1.86(1.07–3.26)	0.64(0.42–0.96)	0.73(0.45–1.20)	1.09(0.72–1.67)
**Team interaction**						
Number of dinners with colleagues						
0–1 time	74.04(72.53–75.55)					
2 times	13.70(12.52–14.88)	1.03(0.73–1.44)	0.80(0.56–1.14)	0.71(0.56–0.90)	0.62(0.47–0.82)	0.65(0.50–0.82)
3 times	5.16(4.40–5.92)	1.48(0.93–2.37)	1.27(0.80–2.02)	0.68(0.47–0.98)	0.50(0.33–0.74)	0.47(0.32–0.69)
≥4 times	7.10(6.22–7.98)	1.96(1.30–2.96)	2.11(1.39–3.19)	0.71(0.52–0.97)	0.48(0.33–0.70)	0.72(0.52–0.99)
Number of clinical case meetings						
0–1 time	23.57(22.11–25.03)					
2 times	19.58(18.22–20.94)	1.47(1.00–2.18)	1.30(0.88–1.90)	1.21(0.95–1.55)	1.12(0.84–1.50)	1.15(0.89–1.48)
3 times	15.82(14.57–17.07)	1.73(1.16–2.59)	1.36(0.91–2.04)	1.35(1.04–1.76)	1.80(1.30–2.50)	1.16(0.88–1.52)
≥4 times	41.03(39.34–42.72)	1.91(1.36–2.69)	1.42(1.02–1.99)	1.74(1.40–2.16)	1.98(1.51–2.59)	1.27(1.02–1.58)

Control variables: Socio-demographic characteristics; hospitals and departments characteristics; family support; patient behavior.

## Data Availability

The study database is available via e-mail to the corresponding authors: Yuan Liang <liangyuan217@hust.edu.cn>.

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
