# Peer review of "Association of Organizational Behavior with Work Engagement and Work-Home Conflicts of Physician in China"

_ijerph, 2021, doi:10.3390/ijerph18105405_

Round 1

Reviewer 1 Report

I wanted to congratulate the authors for their work, it seems to me that it is very new and well structured. However I have some suggestions to improve the article:

1. The abstract is not in a proper format.

2. The level of confidence and estimation error should be indicated in the sample.

3. The criteria for dividing the binary variables and the categorization should be clear in the text.

Regards

Author Response

please see PDF file.

Reviewer 2 Report

Dear author/s,

Thanks for the opportunity to read and review the article:

“Association of Organizational Behavior with Work Engagement 2 and Work-home Conflicts of Physician in China 3”

The fundamental conclusion of the study is that if there is a causal relationship between work engagement and home-work conflict for the medical profession, the object of intervention should never be the physician's own behavior but that of the health organization in which he/she works.

The object of research seems to me of interest and very current. A national cross-sectional survey of 3.255 physicians between July 2014 and April 2015 has been performed in China.

However, before continuing to delve into the study it is necessary for the authors to explain to me the reason for the following data:

Undergraduate and below 1679 respondents (52.93 %)

Traditional Chinese medicine 904 respondents (27.78 %)

In my opinion, if the subject of the investigation is physicians, they cannot be undergraduates.

If those who practice Traditional Chinese Medicine (904 respondents) are included among the undergraduates, the data do not coincide. In the sample, there are many more undergraduates (1.679 respondents) than TCM practitioners (904 respondents).

On the other hand, physicians are characterized, according to the authors, as follows:

Internal medicine 1.723 respondents (52.94 %)

Surgery 1.532 respondents (47.06 %)

This indicates that all respondents are physicians, and therefore should not be undergraduates.

Perhaps the problem is the nomenclature, that is, who are the physicians in China?

Please, I need to know who exactly the physicians are, why is there a majority of undergraduates, and their distribution in internal medicine and surgery, that is, what kind of jobs are included in the undergraduate respondents of internal medicine and surgery.

Thanks for giving me this explanation.

Author Response

Please see PDF file.

Reviewer 3 Report

Dear researchers, I have read your paper that describes a very interesting topic. But your manuscript presents many weak points. Here are some of them:   

1) You have to improve the abstract. It is not easily understood

2) Line 45-62, please present some references

3) you have to describe the theoretical and scientific state of the art concerning the variables under study. It is a very important weak point.

4) What are the hypotheses of your research?

5) you should describe why you have adopted the specific measures.

6) your research uses a cross sectional survey, how you have analyzed/managed the Common Method Biases. 

7) binary logistic regression doesn't study a general association among variables, but the effect of the predictors (independent variables) on a binary predicted variable (the dependent variable).

8) You have to describe your findings following the classic scientific standard, so your table and figure present redundant information (mot focused on the aims).

9)  The discussion should compare with more details your results with the evidence of the literature.

I think that the weak points of your manuscript are too many for reconsideration of it. In my opinion, the data is interesting but a completely new writing of the paper is needed.

Best Regards

Author Response

Please see PDF file

Reviewer 4 Report

Dear authors.

Thank you very much for your interesting work.

I think it is a very interesting and current topic, however, I would like to offer some suggestions to be considered.

First of all, I found Introduction quite brief. The variables in your study are not explained, and I think that you should explain engagement, and work family conflict. Besides, you are taking organizational justice into account, and its not described in the introduction section. I think that extending introduction would ease the understanding of your research.

I have found data analysis clear and complete. 

Discussion section is clear as well.

To sum up, I have found your manuscript very interesting and well constructed. I would only recommend to complete the introduction section.

Thank you very much.

Author Response

Please see PDF file.

Round 2

Reviewer 2 Report

You have to write a new article based on the previous data. 

Reviewer 3 Report

Dear authors,
I very much appreciated the work you have done on the paper. However, I think that two important points are still missing:
1) in the introduction I asked you to present the state of the art of the variables that you study (organizational fairness, leadership attention, team interaction, Work Engagement, Work-home conflicts); it is not enough to present the state of the art regarding organizational behavior. At least for the two outcomes (Work Engagement, Work-home conflicts) it is necessary to define the construct (what are Work Engagement and, Work-home conflicts?), the predictors (partly done), and the consequences as indicated by other studies in the literature.
2) Finally, in the conclusions it is necessary to indicate whether the hypothesis has been confirmed or not.
